# The Quality of Five Natural, Historical Italian Cheeses Produced in Different Months: Gross Composition, Fat-Soluble Vitamins, Fatty Acids, Total Phenols, Antioxidant Capacity, and Health Index

**DOI:** 10.3390/ani12020199

**Published:** 2022-01-14

**Authors:** Adriana Di Trana, Ambra Rita Di Rosa, Margherita Addis, Myriam Fiori, Antonino Di Grigoli, Valeria Maria Morittu, Anna Antonella Spina, Salvatore Claps, Vincenzo Chiofalo, Giuseppe Licitra, Massimo Todaro

**Affiliations:** 1School of Agricultural, Forestry, Food and Environmental Sciences (SAFE), University of Basilicata, 85100 Potenza, Italy; 2Department of Veterinary Sciences, University of Messina, 98168 Messina, Italy; ambra.dirosa@unime.it; 3AGRIS Agris Sardegna, Loc. Bonassai, 07040 Olmedo, Italy; mfiori@agrisricerca.it; 4Department of Agricultural, Food and Forest Sciences (SAAF), University of Palermo, 90128 Palermo, Italy; antonino.digrigoli@unipa.it (A.D.G.); massimo.todaro@unipa.it (M.T.); 5Department of Health Sciences, University Magna Græcia of Catanzaro, 88100 Catanzaro, Italy; morittu@unicz.it (V.M.M.); aa.spina@unicz.it (A.A.S.); 6CREA Research Centre for Animal Production and Aquaculture, 85051 Bella Muro, Italy; salvatore.claps@crea.gov.it; 7Department of Chemical, Biological, Pharmaceutical and Environmental Sciences, University of Messina, 98166 Messina, Italy; vincenzo.chiofalo@unime.it; 8Department of Agriculture, Food and Environment (Di3A), University of Catania, 95123 Catania, Italy; glicitra@unict.it

**Keywords:** natural historical cheeses, extensive system, cheese quality, chemical composition, fat-soluble vitamins, fatty acids, total phenols, antioxidant capacity, health index

## Abstract

**Simple Summary:**

For the purposes of raising awareness of five historical cheeses of Southern Italy that are less known by consumers, and of restoring dignity to the breeders and producers of these cheeses, we studied their quality in terms of chemical composition, monounsaturated fatty acid (MUFA), polyunsaturated fatty acid (PUFA), conjugated linoleic acid (CLA), PUFA-ω6, PUFA-ω3, α-tocopherol, retinol, cholesterol, polyphenol content (TPC), total antioxidant capacity (FRAP and TEAC), and health index (GHIC). Two stretched-curd bovine cheeses, *Caciocavallo Palermitano* (CP) and *Casizolu del Montiferru* (CdM), two ovine cheeses, *Vastedda della Valle del Belìce* (VVB) and *Pecorino Siciliano* (PS), and one caprine cheese, *Caprino Nicastrese* (CN), were evaluated. These cheeses are produced in different months, with raw milk from animals reared in an extensive feeding system. In April, the CP cheese showed high values for CLA, TPC, and GHIC, while the CN cheese exhibited high PUFA, PUFA-ω6, PUFA-ω3, TEAC, and GHIC. In May, the CdM cheese exhibited high content of fat, saturated fatty acids, PUFA-ω3, α-tocopherol, TEAC, and GHIC, while the PS cheese showed high values of protein, CLA, PUFA, PUFA-ω3, α-tocopherol, and GHIC. These measured parameters characterize and distinguish each cheese due to links with numerous factors: species, breed, feeding system, pasture biodiversity, climate, production technology, traditional tools, and ripening type. It is highlighted that, in general, the highest nutritional quality, linked to the highest presence of healthy compounds, originates from the pasture of cheese production in the spring.

**Abstract:**

Five natural historic cheeses of Southern Italy were investigated—*Caciocavallo Palermitano* (CP), *Casizolu del Montiferru* (CdM), *Vastedda della Valle del Belìce* (VVB), *Pecorino Siciliano* (PS), and *Caprino Nicastrese* (CN)—which are produced with raw milk and with traditional techniques and tools, from autochthonous breeds reared under an extensive system. The effects of the month of production on gross composition, MUFA, PUFA, PUFA-ω6, PUFA-ω3, α-tocopherol, retinol, cholesterol, TPC, TEAC, and GHIC were evaluated. In CP, CLA, TPC, and GHIC were higher in April than in February. CdM showed higher values in terms of fat, saturated fatty acids, PUFA-ω3, α-tocopherol, TEAC, and GHIC in May than in February and September, while low values in terms of protein, moisture, and CLA were found. In VVB, MUFA, PUFA-ω6, and α-tocopherol increased in June compared with April; conversely, protein, FRAP, and TEAC were higher in April. In PS, protein, CLA, PUFA, PUFA-ω3, α-tocopherol, and GHIC increased in May compared with January; on the contrary, moisture, NaCl, and TEAC showed high values in January. CN showed higher values in terms of PUFA, PUFA-ω6, PUFA-ω3, TPC, TEAC, and GHIC in April and June compared with January. It is shown that each cheese is unique and closely linked to the production area. Cheeses produced in the spring months showed a high nutritional quality due to the greatest presence of healthy compounds originating from an extensive feeding system.

## 1. Introduction

The Mediterranean dairy sector, characterized by a high biodiversity (animal and vegetal) of systems and dairy products, represents one of the most important activities connected to the utilization of mountain and marginal lands and to the production of typical and quality products. A “typical” product is an out-turn of several factors that are closely related to the geographical origin and to the social and cultural traditions of the production area [1]. The importance of the typicality of dairy products as a tool for enhancing the biodiversity, culture, and economy of vast Mediterranean areas is underlined by Uzun et al. [2]. The European Union officially recognizes this product diversification through the attribution of protected designation of origin (PDO) and protected geographical indication (PGI) quality labels. In Italy’s agro-food policy, the traditional agri-food product (TAP) label is also present.

Recently, 15 natural historic cheeses of Southern Italy, identified as the “AGER (Agroalimentare e ricerca) selection”, received attention in the project entitled “*Canestrum casei*—Development of a Synergy Model Aimed to Qualify and Valorize the Natural Historic Cheese of Southern Italy in the Sicilian, Sardinia, Calabria, Basilicata and Campania Regions”. The project aims to qualify and valorize these historic cheeses and to restore dignity to the breeders and producers of these cheeses, which are less known to consumers and often at risk of extinction due to objective difficulties in terms of their qualification and exploitation in the market. 

In this work, 5 of the 15 cheeses of the AGER selection are studied. 

The *Caciocavallo Palermitano* TAP is a hard cheese, produced in the Sicily region in the province of Palermo at 700–1000 m a.s.l. It is a stretched-curd cheese obtained from the raw bovine milk of Cinisara, Pezzata Rossa, and Bruna breeds and half-breeds, and is still made with traditional techniques (Table 1) using wooden tools. The manufacturing technology of the *Caciocavallo Palermitano* TAP cheese is described by Bonanno et al. [3]. 

The *Vastedda della Valle del Belìce* PDO is a soft stretched-curd cheese, manufactured in the hilly (300 m a.s.l) area of Western Sicily. It is made from raw ovine milk of the autochthonous Valle del Belìce sheep breed. This cheese is made using traditional wooden equipment and applying the stretching technology (Table 1), which gives it uniqueness. Cruciata et al. [4] describe the manufacturing technology of the *Vastedda della Valle del Belìce* PDO cheese. 

The *Pecorino Siciliano* PDO is a hard cheese product from the mountainous area (700 m a.s.l.) of Central Sicily and the hilly area (400 m a.s.l.) of Western Sicily. It is produced with the raw milk of autochthonous sheep breeds—Valle del Belìce, Comisana, and Pinzirita—reared under an extensive system. *Pecorino Siciliano* cheese is made using traditional wooden equipment and technology, as reported briefly in Table 1. The *Pecorino Siciliano* PDO manufacturing technology is described by Todaro et al. [5].

The *Casizolu del Montiferru* TAP is a hard cheese produced in the Sardinia region in the mountainous area (1050 m a.s.l.) of Montiferru and the hilly area (500 m a.s.l.) of Montiferru/Guilcer. It is a stretched-curd cheese obtained from the raw bovine milk of Sardo-Modicana and Bruno-Sarda breeds, reared under an extensive system during the year, using traditional techniques (Table 1) with copper and wood equipment. The manufacturing technology of *Casizolu del Montiferru* TAP is reported by Pinna et al. [6]. 

The *Caprino Nicastrese* is a semi-hard cheese produced in the Calabria region in the hilly (300–600 m a.s.l.) and mountainous (601–1200 m a.s.l.) areas surrounding Nicastro-Lamezia Terme, in the province of Catanzaro. It is made from the raw milk of the autochthonous Nicastrese goat breed, using a traditional technique (Table 1) with wood and steel equipment. 

These five cheeses are produced in restricted areas, pedoclimatic conditions, and anthropic activities (defined as unique and not reproducible elsewhere), which, as well as the territory and habits, are in accordance with history and tradition.

One of the specific objectives of the “*Canestrum Casei*” project was to realize a fact-finding survey on the quality of these cheeses to fulfill the characterization of the AGER cheese selection by standardizing information of greater interest to consumers with respect to regional production specificities. In particular, in the present work, the influence of the production period on the chemical composition, nutritional characteristics, polyphenol content, total antioxidant capacity, and health index of the above-mentioned five traditional cheeses, was evaluated.

## 2. Materials and Methods

### 2.1. Survey Design and the Collection of Cheese Samples from Farms

This study was carried out within the project “*Canestrum casei*”. Five farmhouse cheese varieties, selected among the natural historic cheeses from the south of Italy and representing five types of cheese-making technologies, were studied: *Caciocavallo Palermitano* TAP (CP), *Vastedda della Valle del Belìce* PDO (VVB), *Pecorino Siciliano* PDO (PS), *Casizolu del Montiferru* TAP (CdM), and *Caprino Nicastrese* NB (CN). For each of the 5 cheeses, the most representative cheese producers were selected: 5 producers for CP, 7 for VVB, 6 for PS, 3 for CdM, and 5 for CN, representing (in percentage of total producers), 20% for CP, 71% for VVB, 46% for PS, 75% for CdM, and 85% for CN, respectively. The cheese sample collection was made at the best-selling ripening stage, when most appreciated by consumers, as shown in Table 1. The collected cheese samples were produced in different periods: CP was produced in February and April, VVB was produced in April and June, PS was produced in January and May, CN was produced in April, June, and January, and CdM was produced in May, September, and February. The main features of these cheeses and the production process are reported schematically in Table 1.

### 2.2. Breeding Systems and Diets

CP is produced from the milk of Cinisara/Pezzata and Rossa/Bruna/half-breed cows, reared in an extensive system. In February, they are fed at pasture, with hay and grain supplementation at night. In April, cows are fed mainly at pasture, with a minute amount of concentrates.

VVB is made from the milk of autochthonous Valle del Belìce ewes, maintained under extensive feeding conditions. In April, sheep spend 30% of the grazing time on natural pastures, and 70% on the cultivated grass of Sulla (*Hedysarum coronarium*), while in June, they graze only on dry natural pastures and on residues of wheat threshing without concentrate supplementation.

PS is obtained from the milk of Valle del Belìce, Comisana, and Pinzirita sheep, reared under an extensive system. In January, the animal diet consists of an average of 25% natural pasture, 25% cultivated pasture, 25% hay, and 25% concentrates; in May, it consists of 52% natural pasture, 25% cultivated pasture, and 23% hay, without concentrate supplementation.

CdM is made from the milk of Sardo-Modicana and Bruno-Sarda cows, reared under an extensive system. In May and February, cows graze mainly on natural pasture and on cultivated Berseem clovers (*Trifolium alexandrinum*) and receive a small amount of ryegrass and oat hay; in September, they graze on stubble pasture and on cultivated Berseem clovers with the supplementation of hay and flour of barley, corn, and peas. CN is produced from the milk of autochthonous Nicastrese goats reared under an extensive system. In April and June, goats graze only on natural pasture without feed supplements, while in January they receive a small quantity of alfalfa (*Medicago sativa*), Italian rye grass (*Lolium multiflorum*), or oat (*Avena sativa*) hay from the farm’s fields.

### 2.3. Chemical Composition of Cheeses

The proximate composition of the cheeses (moisture, fat, protein, and NaCl content) was analyzed according to official methods. Moisture was determined on a sample of about 3 g weighed into pre-weighed aluminum dishes with 20 g of sea sand at the bottom. The sample was dried in an oven for 4 h at 102 °C to a constant weight [7]. Total fat content was determined with 3 g of cheese. The sample was weighed in a glass tube, and 8 mL of 25% HCl (aqueous solution *v*/*v*) was added to completely digest the material in a bath of boiling water. The fat was then extracted using a separatory funnel with ethyl ether and petroleum ether as solvents [8]. Protein content was determined by the Kjedahl nitrogen method on about 1 g of cheese directly weighed into the digestion flask and fixed for 1 h in the digestion unit at 450 °C. The digest was then distilled and titrated against 0.1 N HCI [9]. Salt was calculated from sodium content (Na × 2.5) according to EU regulation N° 1169/2011 [10]. About 0.2 g of the sample was weighed into a digestion vessel, and 3 mL of 65% nitric acid (aqueous solution, *v*/*v*) was added to digest the material. The digest was then analyzed for sodium by a Thermo Fisher Scientific ICP/MS [11,12].

### 2.4. Total Retinol, α-Tocopherol, and Cholesterol

The amount of α-tocopherol (Vitamin E), total retinol (Vitamin A), and total cholesterol in the cheese samples was determined by reversed phase HPLC methods proposed by Panfili et al. [13] and Manzi et al. [14]. Briefly, aliquots (0.5 g) of cheese were digested with 2 mL of KOH (60% aqueous solution, *w*/*v*), 2 mL of 95% ethanol, 1 mL of NaCl (1% aqueous solution, *w*/*v*), and 5 mL of an ethanolic solution of pyrogallol (6%, *w*/*v*) added as an antioxidant. After digestion, in a water bath at 70 °C, the suspension was cooled for 30 min, and 5 mL of an NaCl solution (1%, *w*/*v*) was added to prevent emulsification. The suspension was then extracted with 10 mL of n-hexane/ethyl acetate (9:1, *v*/*v*). The lower aqueous layer was extracted 3 more times, with 5 mL of n-hexane/ethyl acetate (9:1, *v*/*v*) each time. The pooled organic layers were evaporated with a rotary evaporator at 30 °C, and the dry sample was dissolved in 3 mL of methanol for HPLC. A sample volume of 20 µL was injected in HPLC equipment, previously filtered using a 0.20 µm PTFE filter. All determinations were carried out in duplicate.

### 2.5. Fatty Acid Profile

Cheese fat extraction was made according to the method of Jiang et al. [15]. Briefly, aliquots (3 g) of cheese were suspended with 10 mL of deionized water and 18 mL of isopropanol. After vigorous shaking, the mixture was supplemented with 13 mL of n-hexane and homogenized using an Ultra–Turrax (T 25 Basic, IKA WERKE, Staufen, Germany) for 3 min at 13,500 rpm. The suspension was then centrifuged (1094× *g*) for 10 min at 4 °C, and the upper organic layer was transferred to a glass test tube. The lower aqueous layer was extracted twice more, with 13 mL of n-hexane each time, and the organic supernatants, after suspension centrifugation, were pooled with the previous hexane layer. The pooled hexane layer was evaporated with a rotary evaporator at 30 °C. The extracted fat was stored at −20 °C until further analysis. 

Fatty acid methyl esters (FAMEs) were obtained from 50 mg of cheese fat trans-methylated according to the ISO 15884/FIL 182 method [16] and then analyzed through GC-FID, as described by Caredda et al. [17]. Individual FAMEs were identified, based both on retention time and on the comparison with a standard mixture of 37 pure components (Supelco 37 Component FAME Mix, Merck Life Science, Milano, S.r.l, Italy). The identification of the isomers of the conjugated linoleic acid (CLA) was accomplished by comparing the retention time of each chromatographic peak and those of a mixture of chromatographic standards (CLA cis9 trans11; CLA trans10 cis12; CLA cis9 cis11; CLA trans9 trans11, Matreya, Restek Italia Superchrom, Milano, S.r.l, Italy). A comparison of the obtained chromatographic profile with those described by Kramer et al. [18,19] was taken into account for confirmatory purposes. The quantitative determination of each FAME was carried out using a calibration curve with internal standards (100 mg of each per g of fat) Me-C5:0 (to quantify FAMEs from C4:0 to C6:0), Me-C9:0 (to quantify FAMEs from C8:0 to C10:0), Me-C13:0 (to quantify FAMEs from C11:0 to C17:0), and Me-C19:0 (to quantify FAMEs from C18:0 to C26:0). Data on the FAMEs were processed to compute the content of saturated fatty acids (SFAs), all cis-monounsaturated fatty acids (MUFAs), all cis-polyunsaturated fatty acids (PUFAs) (as defined in EU Regulation No 1169/2011), PUFA-ω6, and PUFA-ω3. All determinations were carried out in duplicate.

### 2.6. Total Phenol Content and the Antioxidant Capacity

#### 2.6.1. Cheese Extraction

Cheese extract was prepared according to the method of Rashidinejad et al. [20] with slight modifications. Briefly, 0.500 g of cheese was suspended with 25 mL of methanol (95% aqueous solution) containing 1% HCl and homogenized using an Ultra-Turrax homogenizer (T 25 D, IKA WERKE, Staufen, Germany) at 12,000 rpm for 1 min. The suspension was kept at 40 °C for 30 min under gentle stirring (200 rpm). The mixture was cooled and filtered with a cheese cloth, and the residues were washed with 1 mL of the same solvent. The extract was centrifuged at 7000× *g* for 10 min at 4 °C, and the supernatant was kept at −80 °C until further analysis.

#### 2.6.2. Total Phenolic Content

The total polyphenol content of the extract was determined using the Folin–Ciocalteu colorimetric method [21]. Briefly, 100 µL of the diluted extract were mixed with 500 µL of a 0.2 N Folin–Ciocalteu reagent. Afterward, 400 µL of a sodium carbonate solution (7.5% aqueous solution, *w*/*v*) was added to the reaction mixture. The absorbance readings were taken at 765 nm after incubation at room temperature for 1 h. Gallic acid was used as a reference standard, and the results were expressed in milligrams of gallic acid equivalents (GAE) per kilogram of cheese. All determinations were carried out in duplicate.

#### 2.6.3. The Ferric Reducing Antioxidant Power (FRAP) Assay

Measurement of the ferric reducing antioxidant power of extract was carried out based on the procedure of Benzie and Strain [22]. The FRAP reagent was prepared by mixing 300 mM sodium acetate buffer (pH 3.6) and 10 mM TPTZ (2,4,6-Tris(2-pyridyl)-s-triazine) in a 40 mM HCl and 20 mM iron (III) chloride solution in a volume ratio of 10:1:1, and the mixture was warmed to 37 °C in a water bath before use. Afterward, 900 µL of the prepared FRAP reagent were mixed with 30 µL of a diluted sample, and absorbance at 593 nm was recorded after 5 min of incubation at 37 °C. A standard curve was constructed using iron sulfate heptahydrate (FeSO_4_·7H_2_O), and data were expressed as millimoles of FeSO_4_ equivalents per kilogram of cheese. All determinations were carried out in duplicate.

#### 2.6.4. The ABTS Radical Scavenging Activity Assay

The ABTS assay was carried out according to the method of Re et al. [23]. Firstly, the ABTS radical cation (ABTS^•+^) was produced by reacting a 7 mM ABTS stock solution with 2.45 mM potassium persulfate (final concentration), allowing the mixture to stand in the dark at room temperature for 16 h before use. For the assay, the ABTS^•+^ solution was diluted with 5 mM phosphate-buffered saline (PBS), pH 7.4, to an absorbance of 0.700 (±0.02) at 734 nm. A measure of 20 μL of water extract was mixed with 2 mL of a diluted ABTS^•+^ solution, and absorbance was read at 734 nm after incubation at 30 °C for 6 min.

A Trolox (6-hydroxy-2,5,7,8-tetramethychroman-2-carboxylic acid) solution was used to develop a standard curve, and the results were expressed as millimoles of Trolox equivalents (TEAC) per kilogram of cheese. All determinations were carried out in duplicate.

### 2.7. The General Health Index of Cheese (GHIC)

The general health index of cheese (GHIC), introduced by Giorgio et al. [24], compacts the contributions of the components present in the cheese that promote the healthiness of the product into a single value. Basically, the GHIC is used for cheeses obtained from animals fed with fresh forage or pasture and is calculated in the function of various indicators, namely, CLA cis9 trans11, PUFA without CLA, PUFA-ω3, polyphenols, and total antioxidant capacity. CLA cis9 trans11, PUFA, and PUFA-ω3 are already known as health-promoting compounds, but the GHIC also takes into account polyphenols and total antioxidant capacity due to their increasing health interest [25,26]. For each indicator, minimum and maximum benchmarks are defined, allowing the indicators to be scaled into scores between 0 (low health value) and 10 (high health value). Afterward, the scores of the different indicators are summed for each type of cheese, and the sum is the GHIC, as described by Giorgio et al. [24]. 

### 2.8. Statistical Analysis

All statistical analyses were performed using the statistical software package Systat 13 [27]. Data were tested for the distribution of the variables with the Shapiro–Wilk test and analyzed with ANOVA procedure. For each cheese, the model included the month of cheese production as a fixed factor. A Tukey comparison test was used for pair comparisons of the means. Least square means were reported, and differences were considered significant at *p* < 0.05. A tendency was declared at *p* ≤ 0.10. 

## 3. Results

### 3.1. Chemical Composition

Table 2 shows the parameters of the mandatory nutrition declaration as per regulation (EU) N° 1169/2011 [10] and relating to the provision of food information to consumers. In particular, moisture, fat, saturated fat, protein, and NaCl contents (grams per 100 g of cheese) for the 5 cheeses of the AGER selection have been reported, and the effect of the month of production for each cheese was evaluated.

In CP, all parameters were highly stable and comparable between winter (February) and spring (April) production.

CdM was affected by the month of production in several parameters. The highest humidity was found in winter (February) compared with autumn (September), decreasing from 36.2% to 35.5%, respectively. Both months of production showed a moisture significantly higher than that produced in spring (May), with a humidity of 31.4% (*p* < 0.001). Fat, SFA, and protein content were similar between winter and autumn production and significantly different from spring production, with a fat and SFA content lower in February and September and higher in May (*p* < 0.05) and a protein content that, on the contrary, was higher in February and September and lower in May (*p* < 0.05). The salt content seems to be unaffected by the month of production.

VVB showed a similar moisture, fat, saturated fat, and NaCl content in spring and summer (46.2% vs. 45.0%, 19.7% vs. 21.9%, 13.4% vs. 13.8%, and 1.7% vs. 1.6%, respectively). Only the protein content was affected by the month of production, showing 27.7% in April and 26.1% in June (*p* ≤ 0.1).

PS showed a higher moisture content in winter (30.9%) than in spring (27.7%) (*p* < 0.001), a similar content of fat and SFA in both, and a significantly different protein content in January (29.9%) and May (35.1%) (*p* < 0.05). The salt content was affected by the month of production, with a lower percentage in May (2.3%) than in January (2.6%) (*p* < 0.05).

The gross composition of CN was not affected by the month of production; all chemical parameters, in fact, showed a relatively high variability within each month.

### 3.2. Fatty Acid Profile, Total Retinol, α-Tocopherol, and Cholesterol Content

According to EU regulation N° 1169/2011 [10], the content of the mandatory nutrition declaration can be integrated, with an indication of the quantities of one or more non-mandatory elements, such as monounsaturated (MUFA) and polyunsaturated (PUFA) fatty acids, fibers, starches, polyols, vitamins, and mineral elements. Table 3 shows the content of some non-mandatory parameters (MUFA, PUFA, total retinol (vitamin A), α-tocopherol (vitamin E), and of CLA cis9 trans11, PUFA-ω6, PUFA-ω3, and total cholesterol determined in the five cheeses under consideration. The effect of the period (month) of production on the aforementioned parameters was evaluated for each cheese (Table 3).

CP produced in spring (April) was characterized by a CLA content that was twice that produced in the winter (February) (0.41 vs. 0.21 g/100 g of cheese, *p* < 0.05). MUFA, PUFA, PUFA-ω6, PUFA-ω3, retinol, α-tocopherol, and cholesterol contents were comparable between the spring and winter varieties.

CdM showed a CLA content that, in relation to the month of production, varies significantly in the following order: February > May > September (*p* < 0.001). The winter–spring production (February and May) denotes a significantly higher PUFA-ω3 and retinol content than the autumn cheese (September) (0.24 vs. 0.25 vs. 0.16 g/100 g of cheese and 0.29 vs. 0.27 vs. 0.21 mg/100 g of cheese, respectively) (*p* < 0.05). Spring cheese also had a significantly higher α-tocopherol content than autumn and winter cheese (*p* < 0.05). MUFA, PUFA, and PUFA-ω6 were not affected by the season of production.

VVB produced in June exhibited a significantly higher amount (*p* < 0.001) of MUFA, PUFA-ω6, and α-tocopherol than the cheese produced in the spring (April) (4.1 vs. 2.4, 0.55 vs. 0.33 g/100 g of cheese, and 1.11 vs. 0.58 mg/100 g of cheese, respectively). No difference was found between the months of production for CLA cis9 trans11, PUFA, PUFA-ω3, or total retinol content.

PS showed a significantly higher content of CLA cis9 trans 11, PUFA, and PUFA-ω3 in spring (May) than in winter (January) (0.41 vs. 0.22, 1.29 vs. 0.96, 0.65 vs. 0.33 g/100 g of cheese, respectively) (*p* < 0.05). The α-tocopherol content also tended to be higher in cheese produced in May. MUFA, PUFA-ω6, and retinol content was comparable between the spring and the winter cheeses. 

CN showed a significantly higher content of PUFA and PUFA-ω3 in cheese produced in the spring (April) and summer (June) compared with the winter (January) (1.09 vs. 0.93 vs. 0.63 and 0.45 vs. 0.39 vs. 0.19 g/100 g of cheese, respectively) (*p* < 0.05). PUFA-ω6 content tended to vary significantly in the following order: April > June > January (*p* < 0.10). CLA cis9 trans 11, MUFA, total retinol, and α-tocopherol content were not affected by the month of production.

In all cheeses under consideration, the cholesterol contents were not affected by the production period.

### 3.3. Total Phenol Content and Antioxidant Capacity

In the five cheeses of the AGER selection, the effect of the month of production on TPC, FRAP, and TEAC is shown in Table 4.

The production period of CP showed a significant effect on TPC. A higher TPC content was detected in April than in February (4.65 vs. 3.52 g GAE/kg of cheese; *p* < 0.05). FRAP and TEAC content was not affected by the production period. In fact, the winter (February) and spring (April) FRAP and TEAC values were between 1.84 and 2.00 (mmol FeSO_4_/kg of cheese) and between 46.83 and 52.37 (mmol Trolox/kg of cheese), respectively.

CdM showed a significant higher TEAC content in the spring (May) than in the winter (February) (18.89 vs. 10.34 mmol Trolox/kg of cheese; *p* < 0.05), while the TEAC value in cheese obtained in September (12.81 mmol Trolox/kg of cheese) was intermediate between the spring and winter values and not statistically different from the above-mentioned values. The month of production did not affect the TPC and FRAP content, and their values were between 2.98 and 3.65 g GAE/kg of cheese and between 1.69 and 2.08 mmol FeSO_4_/kg of cheese, respectively.

VVB produced in April, compared with June, was characterized by a significantly higher FRAP (*p* ≤ 0.10) and TEAC (*p* < 0.05) content (2.19 vs. 1.74 mmol FeSO_4_/kg of cheese and 69.47 vs. 47.91 mmol Trolox/kg of cheese, respectively). No difference was found in TPC between the months of production (4.62 vs. 4.95; *p* > 0.10).

PS exhibited a significant higher TEAC content in winter (January) than in spring (May) (52.95 vs. 25.42 mmol Trolox/kg of cheese, respectively). No changes in TPC (4.63 and 4.55 g GAE/kg of cheese) or FRAP (2.93 and 2.64 mmol FeSO_4_/kg of cheese) were observed between months of production.

The TPC content of CN increased from January to June; in particular, in June and April, TPC was higher than in January (4.46 vs. 3.58 vs. 2.30 g GAE/kg of cheese, respectively; *p* < 0.01), and the phenol content in June was higher than it was in April (4.46 vs. 3.58 g GAE/kg of cheese, respectively; *p* < 0.05). Regarding the antioxidant capacity, in April and June, CN was characterized by a significantly higher TEAC content than that observed in January (49.11 vs. 38.33 vs. 6.89 mmol Trolox/kg of cheese, respectively; *p* < 0.001), while the month of production did not affect the FRAP values, ranging from 1.76 to 2.32 mmol FeSO_4_/kg of cheese.

### 3.4. General Health Index of Cheese (GHIC)

The GHIC was significantly affected by the month of production (Figure 1), except for the VVB. In general, a significantly higher GHIC was detected in spring and in summer compared with the winter. Within stretched paste cheeses from bovine milk, the GHIC was higher in CP produced in spring (April) and in CdM produced in spring (May) and winter (February) (*p* < 0.05). Regarding ovine cheeses, PS produced in May showed the highest GHIC (*p* < 0.05). No difference was found between production periods in VVB. CN from caprine milk produced in April and June showed higher GHIC values than in the winter (January) (*p* < 0.01).

## 4. Discussion

### 4.1. Chemical Composition

The chemical profiles of the cheeses presented in this work highlight the uniqueness of traditional cheeses. The mandatory chemical parameters, useful for nutritional declarations, were investigated, and the results showed an expected variability due to the numerous factors involved in the production of these traditional cheeses, in terms of, e.g., species, breed, feeding regimen, soil type, geographical position, and climate [28]. Furthermore, an important role in the chemical profile of the five cheeses under examination is played by the artisanal methods of production and the ripening time. The month of production affected the moisture content in CdM and in PS, and humidity was significantly lower in spring cheeses (May production) in both. It is probable that the long ripening time, which characterizes both cheeses compared with the others investigated (180 days for CdM and 210 for PS), increases the significance of the effect of the production period on moisture. Due to the higher water losses, CdM made in spring showed a greater fat content than those made in autumn and winter. The effect of the month of production and of water loss is also shown by the protein content of PS with the same trend (a higher protein content in spring).

Overall, the results obtained for the parameters investigated (moisture, fat, and protein) are comparable to those obtained by other authors in CP [3,29], CdM [6], VVB [30,31,32], and PS [33,34]. Previous studies reported the gross composition parameters of CN at 60 days of ripening [35,36], while there are no data in the literature on CN with a lower degree of ripening. Since consumers are increasingly aware of the importance of proper nutrition, demands for a wider assortment of high-quality products have increased. One of the advantages of cheese is that it is considered an alternative source of protein, which is extremely important in a vegetarian diet. Furthermore, in comparison to milk, it contains more fat and protein and has a high calcium content, and, during ripening, certain components are separated into simpler substances that are more easily absorbed by the human body [37,38].

The salt content of the different cheeses in our study varied from 1.6% (VVB in June) to 2.6% (PS in January), showing very small variation according to the month of production. Cheese consumption is increasing worldwide, and a major objection to cheese is its higher salt content. Public health policies currently aim to prevent diseases associated with an excessive intake of sodium [39,40,41]. To reduce salt consumption, nutrition labelling may be used as a cost-effective intervention to improve population nutrition [42]. Our investigation, aimed at creating correct nutritional declarations, highlights that traditional cheeses, consumed in the right quantities, can and must be part of a healthy diet.

### 4.2. Fatty Acid Profile, Total Retinol, α-Tocopherol, and Cholesterol Content

PUFA, total retinol, and α-tocopherol are non-mandatory parameters that can be integrated into nutrition declarations, as per EU regulation N° 1169/2011 [10], as well as PUFA-ω6, PUFA-ω3, and CLA cis9 tran11, which are considered as presumed indicators of the nutritional quality of foods. Ruminant dairy products are, for example, major dietary sources of conjugated linoleic acids (CLA)—in particular, the CLA cis9 trans11 isomer. This compound can help to prevent carcinogenesis and atherogenesis [43,44] and favor improvements in blood lipids through the reduction in the ratio of low-density lipoproteins (LDLs) to high-density lipoproteins (HDLs). CLA cis9 trans11 originates from both the ruminal biohydrogenation of dietary linoleic and linolenic acid and from endogenous synthesis in animal tissues [45]. The endogenous synthesis, starting from C18:1 trans11 via the Δ_9_-desaturase enzyme in the mammary gland, is a major source of CLA cis9 trans11 in milk and dairy products [46]. Dietary linoleic and linolenic acids, which escape from ruminal biohydrogenation and reach tissues via the blood stream, compete in a series of desaturation and chain elongation reactions leading to long-chain PUFAs. Linoleic acid is a precursor to the PUFA-ω6 series (e.g., arachidonic acid) and the linolenic acid of the PUFA-ω3 series (eicosapentaenoic acid (EPA), docosapentaenoic acid (DPA), and docosahexaenoic acid (DHA)). The multiple and different effects of PUFA-ω6 and PUFA-ω3 on antiatherogenic and antithrombogenic activity in humans are widely recognized [47]. Nutritional recommendations are to increase the level of PUFA-ω3 fatty acids in the consumer diet by focusing on foods with a PUFA-ω6/PUFA-ω3 ratio not exceeding 4:1 [48].

There are no results in the literature relating to the acidic profile and fat-soluble vitamins (total retinol and α-tocopherol) of CdM and CN. Regarding CP, VVB, and PS, the results obtained in the present work are comparable to those reported by other authors [3,30,49]. As shown in Table 3, in the fatty acid profile of the five cheeses in particular, CLA cis9 trans11 and PUFA-ω3 content differed depending on the production period. CLA content in CP, CdM, and PS and PUFA-ω3 content in CdM, PS, and CN were higher in spring (April and May) than in autumn and winter. This trend is linked to the feeding system to which the animals are subjected, which varies according to the season. Similar results were obtained by Bonanno et al. [3] in CP produced from the milk of cows fed at pasture or with a hay- and concentrate-based feed. As observed by Todaro et al. [30], VVB produced in the summer (June, Table 3), compared with the spring (April, Table 3), has a higher content of MUFA and PUFA-ω6. This trend could be due to the fact that, in June, ewes grazed on dry natural pasture and on residues of wheat threshing, which contains wheat grain rich in unsaturated fatty acids—in particular, linoleic acid (precursor of PUFA-ω6) and oleic acid (C18:1) [50]. As reviewed by different authors, the feeding system plays a major role in modulating the fatty acid composition of cow, goat, and sheep milk [50,51,52,53]. Milk from animals grazing a native pasture with a higher botanical biodiversity, which occurs in the spring in Mediterranean pastures, contains more CLA and PUFA and less saturated fatty acid and shows a lower PUFA-ω6/PUFA-ω3 ratio than milk sourced from animals fed at stalls [54,55,56]. Recently, Cabiddu et al. [57] reported that grass-fed animals, compared with those on a conventional maize silage- and/or grain-based diet, produce milk that is higher in PUFA-ω3, retinol, α-tocopherol, carotenoids, and phenols. This effect is more pronounced if animals have grazed on legume- or forb-rich grasslands. As observed by different authors [54,57], pasture richness in a ruminant diet, compared with a stall feeding system, could also enhance the oxidative stability of milk due to higher levels of lutein, carotene, retinol, and α-tocopherol in milk. In the analyzed cheeses (Table 3), the period of cheese production affected the retinol and α-tocopherol content. Both liposoluble vitamins are higher in spring and late spring with respect to the winter and autumn (retinol in CdM (*p* < 0.05) and α-tocopherol in CdM (*p* < 0.05) and PS (*p* < 0.10)). In VVB, a significantly higher α-tocopherol content was observed in cheese produced in June compared with that produced in April. This aspect is probably due to the presence of wheat in the threshing residues grazed by animals in June, which is rich in α-tocopherol [58]. Liposoluble vitamins play a key role in the prevention of human diseases, and the retinol compound is essential for the development and proper functioning of the immune system, whereas α-tocopherol plays an important role in preventing lipid and cholesterol oxidation.

Concerning the analyzed parameters, generally, regardless of the type of cheese, a high variability in the results, due to the artisanal character of production, is emphasized.

### 4.3. Total Phenol Content and Antioxidant Capacity

Phenolic compounds are a group of phytochemicals with potential health-promoting effects [59] and are widespread in the plant kingdom. Their effectiveness as dietary antioxidants in animal feeding has received considerable attention in the last decade [57,60]. The presence of these compounds in milk and dairy products has been observed as a result of the intake of some forage species [61,62,63], natural pasture [64], and aromatic plants [62,65], as well as free-range grazing [66]; these results suggest that there should be a focus on the possibility of enriching milk and cheese with bioactive molecules via the animal diet. The polyphenolic profile of forage or pasture depends on the botanical species, environmental features, and the phenological stage at harvest or at animal intake [67,68]. In addition, bioavailability in the rumen and gastrointestinal tract is due to several factors (molecular size, chemical structure, the degree of polymerization, interaction with the feed matrix, dose intake, and rumen and gut microbiota action) [69], which act on the degradation and absorption of these compounds or their metabolites [59,70]. This complex mechanism affects the transfer of these bioactive molecules from the animal diet to milk and cheese. 

As mentioned previously, there have, so far, been no studies that report on TPC in these five cheeses. TPC content seems to mark CdM, VVB, and PS, regardless of the production month, while TPC was higher in CP and CN produced in the spring and summer (Table 4) compared with the winter. CN produced in January, April, and June and CP produced in February and April were made from the milk of animals grazing on natural pasture without feed supplements or, in those produced in the winter, receiving hay and a small quantity of concentrates. An increase in TPC in goat cheese obtained from grazing animals, compared with animals subject to an indoor feeding system, was found by Hilario et al. [64]. In goat milk, whey, and cheese, phenolic compounds and antioxidant capacity were affected by the feeding system (free-range grazing vs. permanent confinement) [66]. These results could indicate a close relationship between TPC content and the seasonal variability of the quality and quantity of the pasture ingested, a peculiar aspect of the restricted production areas of these cheeses.

The measurement of the antioxidant capacity (AC) of food, including dairy products, is of growing interest since it may provide useful information, e.g., antioxidant activity can be transferred inside an organism when certain foods are ingested, and this is a novel indicator of diet quality [26]. In a wider study, Carlsen et al. [71] observed, among 3100 foods, that cheese had a slightly higher AC than milk and yoghurt. In the current study, we used a FRAP assay and TEAC (ABTS assay) to determine AC in cheeses. AC depends on a wide range of factors, as pointed out by several authors. The main factors that affect TEAC are caseins, β-carotene, uric acid, vitamin E, phenols, whey protein, and folate (microbial influence) [72]; in addition, the cheese-making process [73] and the type of coagulant [74] generate AC variations. The TEAC value is higher in cheeses produced in the summer compared with cheeses produced in the winter [75] and in ripened cheeses compared with fresh cheeses [76], respectively. A comparison of the data among various studies is difficult due to the different methods used for AC assays. 

In our study, TEAC characterizes each cheese and highlights a different antioxidant capacity depending on the month of production, except for CP. Regarding CdM, although animals used the same feeding program in the spring and winter (April and February), the different TEAC values between cheeses produced in different months could be linked to the quantitative and qualitative availability of grass in pastures during seasonal evolution. The effect of the feeding system at pasture during seasonal evolution on AC (FRAP assay) and TPC was evaluated in milk from cows reared in intensive rotational grazing, semi-intensive conventional grazing, and conventional grazing [77]. In this study, a positive correlation between AC and TPC was observed, and AC changed slightly between seasons. In the CN goat cheese, the observed increase in TEAC from January, April, and June (Table 4) could be related to the different composition of natural pasture, which depends on the season. Lucas et al. [78] observed a wide variability of antioxidant activity (FRAP assay) in Rocamadou goat cheese due to the feeding regimen. VVB produced in April had better TEAC and FRAP conditions than that produced in June. This result can be linked by two factors. The first is the high protein content observed in cheese made in April (Table 2). Protein, especially casein [79], contains antioxidant activity. The second is diet composition. In April, sheep consumed a higher quantity of Sulla (*Hedysarum coronarium*) than in June. Sulla forage is characterized by a high level of polyphenols [68].

On the contrary, PS showed a higher TEAC in January compared with April. This result could be related to higher Na content in (Table 2) cheese produced in January, compared with April. In relation to this observation, Lucas et al. [78] found a positive correlation between AC and Na content in cheese (r = 0.432). Antioxidant capacity (FRAP assay) did not indicate great specificity for antioxidant compound content in these five cheeses (Table 4). From a general point of view, the TEAC characterized the production month. This trend was found in some cheeses, but not in all of them. The most likely explanation of these results is in the seasonal variability of the animal’s diet, which is unique in restricted production areas, and in the synergistic action of antioxidant compounds, peculiar to single cheeses, which distinguishes each of them.

### 4.4. General Health Index of Cheese (GHIC)

The GHIC index combines the health compounds present in cheese and provides prompt information concerning the health value of cheeses produced in different months. It should be noted that the GHIC index must be compared within cheeses produced with milk of the same species. It is known that milk fat and protein, sources of healthy compounds, are closely linked to animal species. Few papers simultaneously evaluate and relate CLA, PUFA, antioxidant capacity, and polyphenol content in cheese and those made under different feeding regimens [23,74,79]. In goat *Caciotta* cheese, produced from the milk of goats fed with different forages, GHIC values were between 10 and 28. These values distinguish cheeses according to the type of forage consumed by goats [24]. The GHIC values observed in CN (15–27) were within the above-mentioned ranges. Comparing the GHICs of bovine stretched-curd cheeses and ovine cheeses, the highest values were observed in CP and PS, respectively.

A future evolution of the GHIC index could involve a widening of the range margins of the parameters considered, so that more types of cheese produced from milk of the same animal species can be evaluated.

## 5. Conclusions

The results showed that nutritional characteristics of the studied traditional cheeses, i.e., the fatty acid content, fat-soluble vitamin and polyphenol content, total antioxidant capacity, and the health index, were affected by the month of production.

In general, the highest nutritional quality was linked to the greatest presence of healthy compounds, which originated from pasture in the spring months of cheese production.

Each cheese is unique and strictly linked to the production area, although a high variability, due to numerous factors—including species, breed, diet, type of soil, geographical position, climate, and production technology—characterizes the parameters analyzed in each cheese.

## Figures and Tables

**Figure 1 animals-12-00199-f001:**
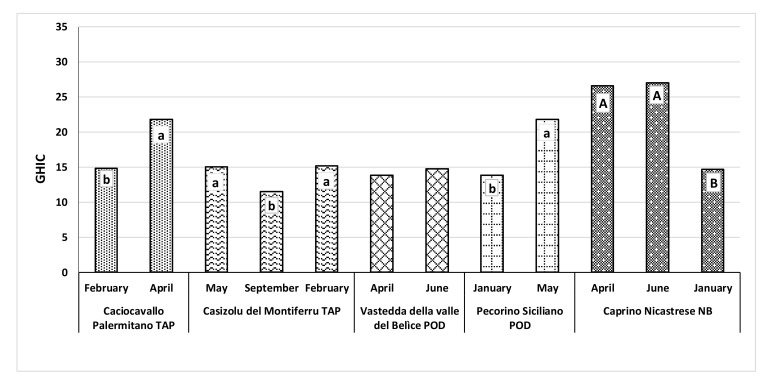
Effect of the month of production on the general health index of cheeses (GHIC) for the five cheeses of the AGER selection. A, B capital letters and a, b small letters within the same cheese type indicate significant differences between months of production, where *p* < 0.01 and *p* < 0.05, respectively.

**Table 1 animals-12-00199-t001:** Synthetic description of general characteristics of the AGER selection cheeses.

Item	Caciocavallo Palermitano TAP	Vastedda della Valle del Belìce PDO	Pecorino Siciliano PDO	Casizolu del Montiferru TAP	Caprino Nicastrese NB
**Region origin**	Sicily	Sicily	Sicily	Sardinia	Calabria
**Production area**	Province of Palermo	Western Sicily	Central and western Sicily	Montiferru	Nicastro-Lamezia Terme-Catanzaro
**Milk nature and species**	Raw bovine	Raw ovine	Raw ovine	Raw bovine	Raw caprine
**Animal breed/s**	Cinisara/Pezzata Rossa/Bruna/ Half-breed	Valle del Belice	Valle del Belìce/Comisana/ Pinzirita	Sardo-Modicana/Bruno-Sarda	Nicastrese
**Curd/Paste**	Stretched/hard	Stretched/soft	Pressed/hard	Stretched/hard	Semi-hard
**Starter**	No	No	No	No	No
**Rennet**	Lamb or Kid, paste	Lamb, paste	Lamb, paste	Calf, liquid	Kid, paste
**Curd breaking**	Seed of chickpea or lentil	Grain of rice	Grain of rice	Hazelnut size	Grain of rice (only in May, as grain of mais or hazelnut)
**Cooking**	For 3 h and, after about 20 h, stretched. In both cases scotta whey is used (85–90 °C)	Stretched with hot (85–90 °C) scotta whey	Under hot (70–75 °C) scotta whey for 3 h	Stretched with hot water (65–75 °C)	No
**Salting**	Brine	Brine	Dry or brine	Brine	Dry
**Shape**	Parallelepiped	Cylindrical	Cylindrical	Pear with head	Cylindrical
**Diameter (cm)**	14 side of the square section	15.17	15–20	13–16	11–13
**Weight (kg)**	~10	0.5–0.7	10	3–3.5	0.8–1.3
**Height (cm)**	38	3	15–25	22–23	5–6
**Ripening (d)**	60	20	210	180	20
**Cheese production (t/year)**	162.5–212.5	40	60	60–65	7.2–9.0
**Total producers (n)**	25	10	13	4	6
**Sampled cheeses (n)**	10	14	11	18	11

Registered trademark PDO = protected designation of origin; TAP = traditional agricultural product; NB = cheese from native breed.

**Table 2 animals-12-00199-t002:** Effect of the month of production on the gross composition of the AGER selection cheeses (g/100 g of cheese).

Cheese Name	Ripening Days	Month of Production	Moisture	Fat	Saturated Fat	Protein	NaCl
**Caciocavallo** **Palermitano TAP**	60	February	39.1	26.4	17.2	27.8	2.2
		April	38.4	26.1	16.3	28.5	2.2
		SEM	0.6	0.8	0.7	0.5	0.1
**Casizolu del** **Montiferru TAP**	180	February	36.2A	27.8b	18.2b	29.9a	2.0
		May	31.4B	29.3a	19.5a	27.4b	2.1
		September	35.5A	27.7b	18.2b	29.9a	2.1
		SEM	0.3	0.4	0.3	0.6	0.01
**Vastedda valle** **del Belìce PDO**	20	April	46.2	19.7	13.4	27.7$	1.7
		June	45.0	21.9	13.8	26.1&	1.6
		SEM	0.7	0.9	0.6	0.6	0.1
**Pecorino** **Siciliano PDO**	210	January	30.9A	30.2	21.0	29.9b	2.6a
		May	27.7B	30.8	20.5	35.1a	2.3b
		SEM	0.5	1.1	0.9	1.1	0.1
**Caprino** **Nicastrese NB**	20	January	42.5	27.0	17.9	21.0	2.3
		April	39.7	26.5	18.1	25.7	2.1
		June	39.8	26.1	17.4	26.0	2.1
		SEM	2.0	1.0	0.9	1.6	0.1

Registered trademark PDO = protected designation of origin; TAP = traditional agricultural product; NB = cheese from native breed; SEM = standard error of mean; a, b small letters, A, B capital letters and $, & symbols in columns within the same cheese type indicate significant differences for *p* < 0.05–*p* < 0.01, *p* < 0.001, and *p* ≤ 0.10, respectively.

**Table 3 animals-12-00199-t003:** Effect of the month of production on the fatty acids (g/100 g of cheese), the liposoluble vitamins, and the cholesterol (mg/100 g of cheese) of the AGER selection cheeses.

Cheese Name	Ripening Days	Month of Production	CLA cis9 trans 11	MUFA	PUFA	PUFA ω6	PUFA ω3	Total Retinol	α-Tocopherol	Cholesterol
**Caciocavallo** **Palermitano TAP**	60	February	0.21b	5.8	0.72	0.50	0.21	0.25	0.60	78
		April	0.41a	5.4	0.79	0.48	0.28	0.26	0.72	86
		SEM	0.05	0.3	0.03	0.03	0.04	0.04	0.10	4
**Casizolu del** **Montiferru TAP**	180	February	0.22A	5.9	0.87	0.63	0.24a	0.29a	0.71b	102
		May	0.20B	5.9	0.85	0.59	0.25a	0.27ab	1.18a	106
		September	0.15C	6.3	0.82	0.66	0.16b	0.21b	0.70b	97
		SEM	0.01	0.1	0.03	0.05	0.02	0.02	0.12	3
**Vastedda valle del Belìce PDO**	20	April	0.26	2.4B	0.86	0.33B	0.50	0.26	0.58B	74
		June	0.25	4.1A	1.03	0.55A	0.46	0.26	1.11A	68
		SEM	0.02	0.2	0.07	0.04	0.04	0.01	0.07	4
**Pecorino** **Siciliano PDO**	210	January	0.22b	4.9	0.96b	0.61	0.33b	0.25	0.36&	86
		May	0.41a	4.3	1.29a	0.61	0.65a	0.27	0.58$	75
		SEM	0.05	0.3	0.10	0.06	0.07	0.03	0.07	6
**Caprino** **Nicastrese NB**	20	January	0.17	5.5	0.63b	0.42£	0.19b	0.41	1.50	104
		April	0.18	4.5	1.09a	0.63$	0.45a	0.29	0.72	95
		June	0.20	5.0	0.93a	0.51&	0.39a	0.32	0.85	103
		SEM	0.03	0.5	0.06	0.05	0.04	0.04	0.27	6

Registered trademark PDO = protected designation of origin; IGP = protected geographical indication; TAP = traditional agricultural product; NB = cheese from native breed; CLA = conjugated linoleic acids; MUFA = monounsaturated fatty acids; PUFA = polyunsaturated fatty acids; SEM = standard error of mean; a, b small letters, A, B, C capital letters and $, &, £ symbols in column within the same cheese type indicate significant differences for *p* < 0.05–*p* < 0.01, *p* < 0.001, and *p* ≤ 0.10, respectively.

**Table 4 animals-12-00199-t004:** Effect of the month of production on the total polyphenol content and the total antioxidant capacity of the AGER selection cheeses.

Cheese Name	Ripening Days	Month of Production	TPC (g GAE/kg Cheese)	FRAP (mmol FeSO_4_/kg Cheese)	TEAC (mmol Trolox/kg Cheese)
**Caciocavallo** **Palermitano TAP**	60	February	3.52b	1.84	52.37
		April	4.65a	2.00	46.83
		SEM	0.27	0.08	3.61
**Casizolu del** **Montiferru TAP**	180	February	2.98	2.08	10.34b
		May	3.25	1.97	18.89a
		September	3.65	1.69	12.81ab
		SEM	0.29	0.19	2.09
**Vastedda valle del** **Belìce PDO**	20	April	4.62	2.19$	69.47a
		June	4.95	1.74&	47.91b
		SEM	0.20	0.17	3.08
**Pecorino** **Siciliano PDO**	210	January	4.63	2.93	52.95a
		May	4.55	2.64	25.42b
		SEM	0.16	0.24	1.82
**Caprino** **Nicastrese NB**	20	January	2.30B	2.32	6.89B
		April	3.58Ab	1.76	49.11A
		June	4.46Aa	1.77	38.33A
		SEM	0.18	0.31	3.57

Registered trademark POD = protected designation of origin; TAP = traditional agricultural product; NB = cheese from native breed; TPC = total phenolic content; GAE = gallic acid equivalent; FRAP = ferric reducing ability power; TEAC = Trolox equivalent antioxidant capacity; SEM = standard error of mean; a, b small letters, A, B capital letters, and $, & symbols in column within the same cheese type indicate significant differences for *p* < 0.05 and *p* < 0.01, *p* < 0.001, and *p* < 0.10, respectively.

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
