# Peer review of "The Quality of Five Natural, Historical Italian Cheeses Produced in Different Months: Gross Composition, Fat-Soluble Vitamins, Fatty Acids, Total Phenols, Antioxidant Capacity, and Health Index"

_animals, 2022, doi:10.3390/ani12020199_

Round 1

Reviewer 1 Report

The works of this type are extremely interesting, and necessary to enhance the traditional products, although, according to many, they are not innovative. Congratulations on your effort.

Reviewer 2 Report

The aim of the research was to investigate five natural, historical cheeses from southern Italy, which are made from raw milk and using traditional techniques and tools, from indigenous breeds kept in an extensive system. The conclusions of the conducted research are clear and result from the obtained research results. The material used for the research is sufficient, the research methods have been selected appropriately. The arrangement of the figure and tabels is clear and presents the obtained results very well. Discussing the results against the background of other authors is very detailed. The publications cited by the authors of the article are well selected. For the most part, the authors refer to the latest knowledge published in renowned scientific journals. I could not find any mistakes in the scientific aspect of the manuscript.

I congratulate the authors for their hard work and nice presentation of their work.

However, the authors did not avoid a few mistakes, which I will list below:

- A few punctuation problems are present in the manuscript. I suggest the Authors to double-check the text.

-In the materials and methods section, please include information about the test used to examine the distribution of variables

Reviewer 3 Report

Manuscript ID: animals-1515647

The Quality of Five Natural, Historical Italian Cheeses Produced in Different Months: Gross Composition, Fat-Soluble Vitamins, Fatty Acids, Total Phenols, Antioxidant Capacity, and Health Index

General remarks

Dear authors,

I have revised the abovementioned manuscript. I very appreciated your manuscript, of which I fully share the aim of promoting the dairy production of southern Italy and, more generally, typical dairy products. The typical production chains promotion is, in my opinion, an effective tool for centuries-old agricultural cultures preservation and, at the same time, for the maintenance of the economy of many marginal areas threatened by abandonment and depopulation. Your contribution, therefore, is greatly appreciated. About the manuscript, the analytical methods are very well described and detailed. Similarly, the discussions are well-argued. In my opinion, however, there are some critical issues regarding the sampling procedures, which could be improved. My comments are listed below. Hoping to have contributed to improving the manuscript quality. Regards

Specific comments

L 35: in my opinion, it is advisable not to start the sentence with an acronym. Thanks.

L 36-38: in my opinion the sentence is unclear. It is not well understood, indeed, to which of the cheeses studied the higher values of the listed parameters refer. I suppose that the values referring to the single cheeses are separated by a semicolon; however, I don't think this form is particularly effective. Authors are kindly requested to rearrange the sentence with a view to greater clarity. Thanks.

L 43 (and along with the study): I would suggest that the authors avoid the concept of "first time". In my opinion, having approached a study for the first time is not a reason for emphasis. On the other hand, research has already been carried out on some of the cheeses studied (the authors provide several references in this regard). Thanks.

L 63: to the biodiversity of which system do the authors refer? Of agro-ecological systems or animal systems? I think it might be useful to point out, thanks.

L 65-67: the importance of the typicality of dairy products as a tool for enhancing the biodiversity, culture, and economy of vast Mediterranean areas is also well underlined by https://doi.org/10.1111/1471-0307.12640, to which the authors are invited to refer. Thanks.

L 72: Is the term "Ager" a Latinism? Many of the readers may miss the point. Please clarify the meaning. Thanks.

L 79: If there is a criterion based on which five of the fifteen cheese varieties included in the project have been chosen, I think it is appropriate to mention it (also in the materials and methods). Thanks.

L 120: the project area to which the manuscript refers has already been specified previously. In my opinion, redundancy should be avoided. Thanks.

L 124-128: I believe that some elucidations are needed. First, it would be useful to know how many product units (Kg of cheese, cheese wheels, and so on) were sampled for cheeses analyses. The number of samples was established on the basis of the type of cheese or the month of production (..... cheese samples, collected according to the type of cheese and the month of production .....)? If so, based on what criteria? Furthermore, do the percentages indicated (20% for CP, 71% for VVB, 46% for PS, 75% for CdM, and 85% for CN) refer to the produced units (Kg of cheese produced, number of cheese wheels) of each individual cheese or to the number of cheese production plants? Are there any statistics to support the production data (e.g., the total number of producers, units of cheese produced)? Based on which criteria were the representativeness of the selected producers established? To what factors can be attributed to the variations between cheeses in the number of surveyed dairies? Authors are requested to clarify these aspects, providing more details if possible. Thanks.

L 129: I am not quite clear if the reported ripening stage (see table 1) corresponds to the usual commercial one or, more simply, to that at which the cheeses studied were sampled. Furthermore, since these are niche dairy products, is there evidence that the indicated ripening stages are the most appreciated by consumers? If possible, the authors should provide more information to support this claim. Thanks.

L 274: in my opinion, it would be appropriate to replace "according to Giorgio et al.'s [23] method" with "as described by Giorgio et al.". Thanks.

L 279 (and along with the text): according to the journal’ template, authors are invited to report the P-value in lower case and italic. Thanks.

L 428 (and along with the discussion): in my opinion, the authors should avoid reporting again the results in the discussion section. Thanks.

L 495: the effect of seasonal variations on the fatty acids profile of milk and cheese due to animal feeding (grazing of Mediterranean native pastures vs winter indoor feeding) was also explored by https://doi.org/10.3390/foods9081091 with regard to ewes, to which the authors are invited to refer. Thanks.

Round 2

Reviewer 3 Report

Dear authors,
 I have read the revised version of the manuscript (animals-1515647). As I mentioned earlier, I consider the manuscript a valuable contribution to safeguarding the "dairy culture" of the inland areas of southern Italy. In light of the changes made, which have dispelled all my doubts, I believe that the current version of the manuscript deserves to be published in the present form. Congratulations. 
Thanks for your patience. Good work and a happy new year. 

Regards